# Induced Pluripotent Stem Cells in Drug Discovery and Neurodegenerative Disease Modelling

**DOI:** 10.3390/ijms25042392

**Published:** 2024-02-18

**Authors:** Daniela Gois Beghini, Tais Hanae Kasai-Brunswick, Andrea Henriques-Pons

**Affiliations:** 1Laboratório de Inovações em Terapias, Ensino e Bioprodutos, Instituto Oswaldo Cruz, Fundação Oswaldo Cruz, Rio de Janeiro 21040-900, RJ, Brazil; andreah@ioc.fiocruz.br; 2Centro Nacional de Biologia Estrutural e Bioimagem, CENABIO, Universidade Federal do Rio de Janeiro, Seropédica 23890-000, RJ, Brazil; tais@cenabio.ufrj.br; 3Instituto Nacional de Ciência e Tecnologia em Medicina Regenerativa, INCT-REGENERA, Universidade Federal do Rio de Janeiro, Seropédica 23890-000, RJ, Brazil

**Keywords:** induced pluripotent stem cells, disease modelling, neurodegenerative diseases

## Abstract

Induced pluripotent stem cells (iPSCs) are derived from reprogrammed adult somatic cells. These adult cells are manipulated in vitro to express genes and factors essential for acquiring and maintaining embryonic stem cell (ESC) properties. This technology is widely applied in many fields, and much attention has been given to developing iPSC-based disease models to validate drug discovery platforms and study the pathophysiological molecular processes underlying disease onset. Especially in neurological diseases, there is a great need for iPSC-based technological research, as these cells can be obtained from each patient and carry the individual’s bulk of genetic mutations and unique properties. Moreover, iPSCs can differentiate into multiple cell types. These are essential characteristics, since the study of neurological diseases is affected by the limited access to injury sites, the need for in vitro models composed of various cell types, the complexity of reproducing the brain’s anatomy, the challenges of postmortem cell culture, and ethical issues. Neurodegenerative diseases strongly impact global health due to their high incidence, symptom severity, and lack of effective therapies. Recently, analyses using disease specific, iPSC-based models confirmed the efficacy of these models for testing multiple drugs. This review summarizes the advances in iPSC technology used in disease modelling and drug testing, with a primary focus on neurodegenerative diseases, including Parkinson’s and Alzheimer’s diseases.

## 1. Introduction

### 1.1. Basic Concepts

Neurodegenerative diseases are characterized by the progressive loss of neuronal and glial populations. They are classified according to their clinical features, such as the anatomic distribution of tissue degeneration and/or primary molecular abnormalities. Among the neurodegenerative diseases, Parkinson’s disease (PD), Alzheimer’s disease (AD), multiple sclerosis, amyotrophic lateral sclerosis, and Huntington’s disease lead to significant social and economic burdens due to their high incidence, symptom severity, and lack of effective therapies. Moreover, their incidence in developed countries is increasing, at least partially because of the proportional increase in the elderly population. The number of individuals with PD may double by 2030 compared to that in 2005 [1], and another study estimated that the number of amyotrophic lateral sclerosis cases worldwide would increase by 69% from 2015 through 2040 [2]. The pathogenic mechanisms possibly involved in this increase are not fully understood, but ageing, genetic propensity, increased protein misfolding, and apoptosis of neural cells have all been implicated [3].

Due to the high complexity of neurodegenerative diseases, no in vivo or in vitro models fully reproduce their phenotype. In the case of animal models, which are more suitable for experimental studies, such as mice or rats, there are significant physiological differences, making it challenging to determine the aetiology and symptoms of these diseases. Moreover, conventional approaches for studying neuronal diseases based on patient brain tissues require samples that can only be obtained postmortem. Although no model for neuronal disease study is perfect, induced pluripotent stem cell (iPSC)-based technology complements and improves upon previous models. Thus, iPSC and gene editing technologies make it possible to use somatic cells from the patient, such as skin or blood, to produce a disease model set in the patient’s genetic context.

The brain comprises neurons and glial cells, which are scaffold cells divided into three types: oligodendrocytes, microglia, and astrocytes. All of these cells are important for optimizing brain function. Oligodendrocytes tightly surround neuronal axons to form the myelin sheath. Microglia play a role in the development and maintenance of neuronal networks and in injury repair. They also have immune properties, as they can detect damage-associated molecular pattern (PAMP) components from damaged neurons and produce inflammatory mediators. Astrocytes are star-shaped glial cells that anatomically provide stromal support to neurons. They can also release gliotransmitters, such as glutamate, to send signals to neighbouring neurons. In addition, through their end-feet connections, astrocytes expand or constrict blood vessels, controlling the flow of nutrients and oxygen to the brain [4].

Embryonic stem cells (ESCs) exhibit pluripotent plasticity and can differentiate into brain cells or into any specialized cell in the human body, leading to the generation of drug test platforms. However, using these cells involves ethical issues [5]. Thus, several milestones have revolutionized the field, allowing the discovery of cellular reprogramming to generate induced pluripotent cells. Gurdon et al., followed by Wilmut et al., showed that the cytoplasm of oocytes contains factors capable of reprogramming somatic cells to a pluripotent stage, capable of generating complete individuals with normal postnatal development. Another paradigm was proposed by Gehring et al. and Davis et al., who presented the concept of master regulatory genes, whose expression modulation can completely convert one cell type into another [6,7,8,9,10,11,12]. In addition, the technical isolation and culture conditions of ESC were defined, and this set of remarkable breakthroughs allowed iPSCs to emerge. iPSCs are generated by the induced expression of the reprogramming transcription factors OCT3/4, SOX2, KLF4, and c-MYC, constituting the first reprogramming cocktail [13]. The first in vitro iPSC reprogramming method was developed by Takahashi and Yamanaka, and this method has improved upon previous methods in terms of efficiency and yield [14,15,16]. iPSCs can differentiate into any cell type in the human body.

Currently, coupled with improvements in reprogramming techniques, this technology has contributed to advancements in the understanding of multiple disease pathologies and assisted the development of more effective therapeutic methods and advances in regenerative medicine [17]. The advent of iPSCs has made it possible to generate any cells of interest from the patient’s somatic cells and thus develop patient-specific drug testing models. Reprogramming somatic human cells from patients with neurodegenerative diseases and healthy individuals (as controls) into human iPSCs and subsequently differentiating these cells into various brain cell types has provided an unprecedented opportunity to study disease mechanisms. Before this advance, transgenic animal models made clinical correlation difficult because of the primary difference between species. A more detailed study of diseases that affect the human brain is hampered because of the invasive and challenging procedures required to obtain living brain material suitable for cell culture. Moreover, the ability to generate neural cultures from postmortem human brains significantly depends on brain tissue quality [18].

In terms of applicability, iPSCs are particularly important because they can be generated from somatic cells obtained by minimally invasive procedures and from any patient. Moreover, these cells potentially reproduce the cellular mechanisms of diseases in vitro, can be differentiated into transplantable cells and tissues compatible with the donor, and can be used to produce cellular platforms suitable for drug testing and screening. Another advantage is that iPSCs carry all naturally occurring genetic mutations and characteristics of individuals, which may affect the pathogenic outcome. In this case, unique pathogenic genetic mutations can be corrected using CRISPR/Cas9, for example, to produce healthy isogenic cells or organoids [19,20].

Constructing multiple levels of neural circuits composed of various cell types is essential for better reproduction of functional, molecular, and cellular responses. For example, complex structures modelling an individual’s genetic characteristics can be produced using 3D bioprinters. In this case, iPSC-derived, terminally differentiated cell types could be anatomically organized into printed organoids [21]. Organoids are miniaturized three-dimensional structures containing multiple cell types and more accurately represent analogues of human organs or tissues. Monolayer cell subtype analysis provides an excellent tool for studying lineage-specific disease mechanisms. However, diseases that are influenced by both the environment and genetics are best studied in multilineage or multisystem studies. Multiple analyses or integrated platforms offer many advantages over monolayer methods, such as microfluidic organs, body chips, organoids, assembloids, 2D cocultures, tissue engineering, bioprinting, chimaeras, and humanized animals [22].

Over the last decade, 3D organoid technology has become more prevalent in stem cell research, and human brain organoids derived from iPSCs have been increasingly used in neurological disease modelling and therapeutic discovery and tests [23]. These 3D brain organoids display critical features of the brain-specific cytoarchitecture and network properties that can be used to study complex neural network phenomena in neurodegenerative disease models. These organoids have been generated to model various brain regions, including the forebrain, midbrain, cerebellum, cortex, and hippocampus [24,25]. In general, to generate these cerebral organoids, two methodologies can be applied: guided and unguided methods (for a review, see [26]). Unguided methods rely entirely on spontaneous morphogenesis methods and take advantage of the intrinsic signalling and self-organization capabilities of iPSCs to spontaneously differentiate into tissues that mimic the developing brain. In this method, the resulting organoids contain heterogeneous tissues that resemble various regions of the brain. In the case of the guided approach, small molecules and growth factors are used to generate spheroids that are specifically representative of a type of tissue. In this way, the guided method can be used to generate two or more organoids representative of different regions of the brain, which can be fused to form assembloids that can model the interactions between different brain regions. Depending on the disease, different modelling methods, strategies, and cell types can be used, allowing the use of selected compounds for drug development and testing. Furthermore, iPSCs offer a nearly unlimited source of human cells that can be made available in public repositories and shared between laboratories and health units [27].

### 1.2. Neurodegenerative Disease Modelling by iPSC Technology

To model neurological disorders in healthy individuals, somatic cells (blood, skin, etc.) are reprogrammed in vitro into colonies of iPSCs. At this stage, genome-editing techniques allow researchers to create additional isogenic cell lines containing specific pathological mutations or transgenes that reproduce a given disease etiology. Conversely, pathogenic genetic alterations can be corrected to generate control iPSC lines when the cells are obtained from patients with neurodegenerative disorders [28,29,30]. Then, the iPSC lines of interest are induced to differentiate into neural cells, including neurons, glial cells, and neural progenitor cells. At the iPSC stage, self-organizing tissue cytosystems or organoids can also be created in three-dimensional culture, not necessarily by using 3D printers (Figure 1).

In Table 1, we show the iPSC lines generated to study the main neurodegenerative diseases. In the case of a specific neurological disease, in AD, for example, neural cells differentiated from iPSCs with a familial AD background can exhibit several AD-like phenotypes that can be tested in vitro. These phenotypes include, for example, amyloid-β peptide production and, for three-dimensional culture, tau pathology, amyloid plaques and synaptic dysfunction. Potential therapeutic small molecules or alternative treatments can be tested directly in human neural cells.

Several attempts have been made to direct pluripotent ESCs towards a culture of neural cells harbouring pathogenic mutations. It was then observed that the stem cells cultured under standard conditions containing TGF-β-related negative inducers and without specific factors to maintain pluripotency showed no induced neural differentiation [59]. Then, specific TGF-β antagonists were used to prevent Suppressor of Mothers Against Decapentaplegic (SMAD) transcription factor family signalling, and robust and significant improvements in differentiation were observed. However, these methods do not recreate human neurogenesis and have important limitations in disease modelling.

Improvements were made, and iPSC differentiation into the cerebral cortex was recapitulated in vivo, leading to the generation of all cortical projections of neurons in a predetermined temporal order. This procedure enabled functional studies about the development of the human cerebral cortex and the generation of ex vivo individual-specific cortical networks for disease modelling [60]. There have also been improvements in the methodology for generating electrophysiologically active neurons without the need for coculture with astrocytes or specialized media [61,62].

Generating isogenic paired cell lines is usually important in iPSC-based disease modelling. These paired control cell lines consist of introducing a given mutation of interest into normal iPSCs or correcting a disease-associated genetic modification to generate a cognate normal lineage. This approach is one of the best for assessing the biological effects of one or several disease-associated mutations. CRISPR-Cas9-mediated gene editing is an excellent tool for modelling monogenic diseases or studying the contribution of single or several gene variants associated with a given pathology [62] (Figure 2).

Obtaining subtypes of neurons, such as dopaminergic and motor neurons, is necessary to model neurodegenerative diseases. Motor neurons are important in modelling diseases such as amyotrophic lateral sclerosis (ALS), and several protocols have been established to generate these neurons from iPSCs [63,64,65,66]. Although ALS is a very complex disease, identifying how motor neurons originate from iPSCs may help to reveal how genes selectively impact motor neuron biology and whether they rely on common pathways to cause neuronal degeneration [48]. Dopaminergic neurons have already been generated from iPSCs and used in the study of PD [67]. A preclinical study using a primate model of PD indicated that human iPSC-derived dopaminergic progenitors were clinically applicable for treating patients [68]. Other neuron subtypes have also been developed and could be used in disease modelling and cell therapy [69].

iPSCs can also be used to generate astrocytes via an intermediate neural progenitor [70]. This cell type is very important in neurological and psychiatric diseases. Jones et al. reported the development of a human iPSC-derived astrocyte model created from healthy subjects and patients with early-onset familial AD or the late-onset sporadic for [41]. These astrocytes can reproduce several phenotypes found in vivo, representing features that could be employed for effective disease modelling [71]. In this context, astrocytes derived from AD patients exhibit a typical pathological phenotype, with a less complex morphological appearance and abnormal localization of key functional astroglial markers [41].

Oligodendrocytes and their precursors are responsible not only for the generation of myelin in the central nervous system but also for the metabolic support of neurons and they have a critical trophic function [72]. Several groups developed protocols to obtain these cells from iPSCs, which were initially based on studies using ESCs [73,74]. Then, an optimized and specific protocol was developed for obtaining oligodendrocytos from iPSCs. This protocol consists of seven steps with an average duration of 150 days, resulting in myelinogenic oligodendrocytes [75]. In a recent study, oligodendrocyte precursor cells derived from iPSCs were grafted into neonatal myelin-deficient shiverer mice, which induced robust brain myelination and substantially increased survival [75]. The authors then improved the protocol for obtaining these cells and reduced the time required for cell differentiation [76]. Following the same strategy, other authors showed the applicability of stem cell-derived oligodendrocytes, which led to remyelination and rescue in irradiated rats, suggesting that brain tumour radiation therapy has excellent therapeutic relevance [77].

Microglia reside in the central nervous system and play essential roles in the development and homeostasis of various neurological and psychiatric diseases. Human iPSCs were differentiated into microglia-like cells by exposure to multiple factors, such as IL-34, TGFβ-1, and CX3CL1, and cocultured with astrocytes [78,79]. It was also shown that human microglia-like cells derived from iPSCs migrated and secreted cytokines in response to inflammatory stimuli. Moreover, they robustly phagocytose central nervous system substrates, including amyloid-β (Aβ) fibrils, brain-derived tau oligomers, and human synaptosomes, similar to conventional microglia [80]. These cells were also used to study the effects of Aβ fibrils and brain-derived tau oligomers on AD-related gene expression. Moreover, they can be used to study the mechanisms involved in synaptic pruning [79]. Microglial-like cells derived from iPSCs are similar to conventional microglia at the transcriptome level and respond to inflammatory stimuli [79]. Figure 3 shows the most relevant neural cell types for modelling the main neurodegenerative diseases.

3D brain organoids derived from iPSCs can partially recapitulate the rearrangement of brain cytoarchitecture, an essential feature for studying the pathogenesis of brain diseases. Lancaster et al. pioneered the development of brain organoids from human pluripotent stem cells. The authors modelled microcephaly, a complex disease that can be reproduced in mice [81]. After this study, other groups used the same approach to generate brain organoids, which helped to model different neurological disorders. These studies using patient-derived brain organoids revealed novel insights into the molecular and genetic mechanisms involved in microcephaly, autism, and AD [82]. A simplified and fast protocol was described for brain organoid induction from human iPSCs [83]. Recently, the generation of arcuate organoids from human iPSCs was shown to model the development of the human hypothalamic arcuate nucleus. Since the existing organoid models do not apply to fine brain subregions, such as different nuclei in the hypothalamus [84], the use of organ-specific progenitor cells highlights the potential of iPSCs in fields other than regenerative medicine.

For most neurodegenerative diseases, age is a common and important risk factor. However, reprogramming somatic cells to iPSCs resets their identity back to embryonic age, and it has been shown that these cells have elongated telomeres and a mitochondrial network with great fitness [85,86]. This, however, diminishes some of the benefits of patient-derived models. To overcome this problem, one approach could involve taking advantage of the biology of known disorders of human premature ageing and engineering disease-associated mutations or overexpressing genes known to cause progeria [87]. In this context, the authors induced the expression of progerin, a truncated form of lamin A associated with premature ageing, and managed to promote ageing in iPSC-derived dopaminergic neurons [88]. Another strategy used to obtain motor neurons from ALS patients is direct reprogramming from fibroblasts [89]. That is, the intermediate pluripotent state can be bypassed by directly reprogramming differentiated cells into neurons. The results of this study revealed that directly reprogrammed motor neurons, rather than iPSC-derived motor neurons, maintained the ageing hallmarks of old donors, including extensive DNA damage, loss of heterochromatin and nuclear organization, and increased SA-β-Gal activity. According to Grenier et al. [87], the use of toxins that induce ROS and/or mitochondrial damage might represent a more versatile and complementary approach to promote ageing in organoids. In fact, the authors used iPSC-derived retinal pigment epithelium cells to model chronic oxidative stress in vitro. For this purpose, paraquat, a known mitochondrial complex I toxin that promotes the formation of ROS, was used to induce chronic stress in the retinal pigment epithelium and to model age-related macular degeneration [90].

### 1.3. Drug Testing in Neurodegenerative Diseases Using iPSCs

One of the prerequisites for drug screening using iPSCs is targeting a relevant cellular phenotype in a given disease. In the first reports of drug screening using iPSCs, neural crest precursors derived from iPSCs were generated from individuals with familial dysautonomia. This disease is a rare and fatal genetic disorder affecting neural crest lineages. It is caused by mutations in the gene encoding the IkB kinase complex-associated protein (IKBKAP), resulting in a splicing defect and a dysfunctional truncated protein. In that work, 6912 small compounds were tested, one of which, known as SKF-86466, was found to improve disease-specific aberrant splicing [91]. In another study involving drug screening, iPSCs derived from patients with sporadic amyotrophic lateral sclerosis and healthy individuals were used and differentiated into motor neurons. The authors de novo identified aggregation of TAR DNA-binding protein 43 (TDP-43) in the patients’ motor neurons. Using a high-content drug screen, they found a compound that reduced TDP-43 aggregation [46]. Other authors used a patient-derived model of iPSCs with spinal muscular atrophy (SMA) to validate specific drugs, and the hit compound was further evaluated in a mouse model. Administration of this compound to mice led to increased survival of motor neurons, higher SMN protein levels, motor function improvement, and neuromuscular circuit protection [92]. This inherited motor neuron disease is caused by a deficiency in SMN expression and results in severe muscle weakness.

Another application consists of drug repositioning using disease-specific iPSCs. In this case, drugs already approved for specific diseases are tested to find new applications for other conditions. One example of this approach showed that iPSC-derived motor neurons produced from amyotrophic lateral sclerosis patients harbouring SOD1 (superoxide dismutase 1) mutations displayed a reproducible, disease-related phenotype and reduced delayed-rectifier potassium channel activity [93,94]. New iPSC-based evidence has shown that correcting motor neuron physiology using the already approved antiepileptic ezogabine drug, a Kv7 potassium channel agonist, reduces neuronal excitability and improves cell survival [93,94]. Drug discovery using iPSCs from patients with multiple genetic forms of a neurodegenerative disease is highly valuable because it allows for the testing of drug responsiveness in multiple patients.

### 1.4. Advances in Specific Conditions Using iPSCs

#### 1.4.1. Parkinson’s Disease

PD is the second most common degenerative disease and affects 2 to 3% of the population older than 65 years. Age is the most important risk factor for developing PD, and men are more susceptible to PD than women, with a prevalence ratio of approximately 3:2. The pathological hallmark of PD is neuronal loss in the substantia nigra, which causes dopamine deficiency and intracellular inclusions containing α-synuclein aggregates [95]. This protein is encoded by the *SNCA* gene, whose duplications or triplications are associated with familial PD. Resting tremors, rigidity, akinesia, and postural reflex disturbance are all tetralogies of PD [96]. PD treatment using cell-based therapies began in the late 1970s and early 1980s when several groups showed that dopaminergic neurons harvested from the developing foetal midbrain could survive in grafts transplanted into PD animal models [97]. It was also revealed that grafted cells could restore brain functionality in a PD rat model [97]. Thus, several studies have been performed using neural cells from different origins, and subsequent studies of dopaminergic neuron transplantation from iPSCs have emerged as a therapeutic option for PD patients. In 2016, the International Stem Cell Corporation started the first approved clinical trial in which iPSCs were used to treat PD patients [98,99].

iPSCs were used to generate dopaminergic neurons, which were obtained from a patient with SNCA triplication and from an unaffected first-degree relative as a control. The patient’s neurons produced twice the amount of α-synuclein protein compared to the unaffected relative, recapitulating the in vitro PD pathology [31]. Several groups have similarly generated dopaminergic neurons from PD patients and healthy controls using iPSCs and compared molecular pathways that might differ. Among these pathways, some are susceptible to therapeutic modulation [72]. For example, iPSC-derived dopamine neurons revealed differences between monozygotic twins discordant for PD. The affected twin’s neurons exhibited a lower dopamine level, increased monoamine oxidase B expression, and impaired intrinsic network activity. Treatment with targeted monoamine oxidase B inhibitors normalized α-synuclein and dopamine levels, confirming the suitability of this system for drug testing [32]. Recently, in another case, autologous dopaminergic neurons were generated from iPSCs obtained from a patient with idiopathic PD. These neurons were implanted back into the patient’s putamen (left hemisphere), followed by right hemisphere implantation after six months. Positron emission tomography (PET) suggested graft survival and clinical control of PD symptoms after surgery, which improved at eighteen to twenty-four months after implantation [33]. This form of cell therapy has shown promising results and is currently one of the best options for slowing or halting PD progression [100]. With advances in cell reprogramming, iPSCs have great potential for treating region-specific neurodegenerations such as PD. In addition, since the cells are patient specific, the chances of an immune response after transplantation are significantly reduced.

#### 1.4.2. Alzheimer Disease

Clinically, AD is the most prevalent cause of dementia and is characterized by memory loss, alterations in personality, and deficits in rational thinking. AD significantly impacts society, as it affects millions of people worldwide, accounting for 60–80% of all patients with dementia. This disease is part of the proteinopathies group and is characterized by amyloid-β (Aβ) peptide deposition as amyloid plaques and tau protein deposition as neurofibrillary tangles [101]. AD is a very complex disease and is still not fully understood. In recent years, a third pathogenic component was shown to be involved in disease onset and progression: the neuroinflammatory response, which is primarily mediated by microglia [102].

Initially, a model of familial AD was established using iPSCs generated from autologous fibroblasts. Mutations in the *PS1* (Presenilin 1), *PS2* (Presenilin 2), and *APP* (amyloid precursor protein) genes account for most familial early-onset cases of AD. Increased production of pathological Aβ leads to a greater tendency to form fibrillary amyloid deposits [34]. According to the experimental model established by the authors, patient-derived differentiated neurons increase Aβ42 secretion, recapitulating the pathological mechanism of familial AD associated with *PS1* and *PS2* mutations. This model was subsequently tested for drugs that could repair genetic mutations. In another work, the authors described the generation of iPSC lines from patients harbouring familial AD based on the *APP* gene mutation (V717I). Significant increases in *APP* expression and levels of Aβ were observed during neural differentiation and maturation. Moreover, an increase in the total level of phosphorylated tau was observed in these genetically manipulated neurons. These studies using human neurons revealed unpredicted effects of the most common familial AD *APP* gene mutation [35]. Other authors used this same model to evaluate therapeutic candidates for AD and tested more than 1000 compounds for their ability to reduce the Aβ load within cultured cells. They obtained 27 promising candidates, and the list was narrowed down to six leading compounds. Afterwards, three candidates were combined to improve the anti-Aβ effect (bromocriptine, cromolyn, and topiramate) as an anti-Aβ cocktail. These results suggested that this iPSC approach could also be used for drug development [36]. Xu et al. used the same technology based on iPSC-derived neurons to examine a chemical library containing hundreds of compounds. Numerous small molecules, including cyclin-dependent kinase 2 inhibitors, can be effective blockers of Aβ1-42 toxicity [37]. This study screened Aβ toxicity using iPSC-derived neurons for the first time, providing an excellent example of how iPSCs can be used for disease modelling and high-throughput compound analysis [37].

Israel et al. tested neurons differentiated from iPSCs obtained from patients with familial and sporadic AD, and the results suggested a direct correlation between the proteolytic processing of the Aβ precursor protein, but not Aβ and glycogen synthase kinase-3β activation or tau phosphorylation, in human neuron-derived iPSCs in culture. This approach allowed us to identify a link between Aβ and tau and additional pathological features of AD [38]. Tau pathology is present in AD and other diseases whose clinical phenotype includes dementia. Iovino et al. showed that neurons derived from iPSCs harbouring a mutation in the *MAPT* gene exhibited abnormal tau expression, tau aggregation and hyperphosphorylation, and multiple disease phenotypes [39], and this system may be useful for drug screening purposes.

Pomeshchik et al., 2020 developed a protocol to generate rapid hippocampal spheroids from human iPSCs, which was subsequently used to model AD. The hippocampus is involved in the formation of new memories, emotions and learning, and is one of the first regions of the brain that atrophy in AD. In that work, the hippocampal spheroids generated from two AD patients harbouring variations in the *APP* or *PS1* gene exhibited cardinal cellular pathological features of AD, including loss of synaptic proteins and an increased ratio of intracellular and extracellular Aβ42/Aβ40 peptides. The authors subsequently developed a gene therapy approach to modulate the expression of genes involved in synaptic transmission. The authors showed that hippocampal spheroids from iPSCs could be used to study the mechanisms underlying early pathogenic changes in the hippocampi of AD patients [40].

In AD animal models, atrophic astrocytes are detected at the earliest stages of the disease, followed by the appearance of hypertrophic reactive astrocytes in response to their proximity to extracellular accumulations of Aβ [103]. The authors also analysed the pathological potential of iPSC-derived astrocytes in AD. In this case, patient-derived induced astrocytes displayed a pathological phenotype, in addition to a significantly less complex morphological appearance and abnormal localization of key functional astroglial markers [41]. The authors reported the development of a human iPSC-derived astrocyte model created from healthy individuals and patients with AD. Astrocyte-derived iPSCs from AD patients exhibited a pronounced pathological phenotype, a general atrophic profile, and abnormal localization of key functional astroglial markers [41]. This work provides a platform for further interrogation of early astroglial cell autonomic events in AD and the possibility of identifying new therapeutic targets for treating this disease. In another study, astrocytes differentiated from AD patient iPSCs showed hallmarks of disease pathology, including increased Aβ production, altered cytokine release, and dysregulated Ca^2+^ homeostasis [42].

Microglia derived from iPSCs have also been used to study their role in neurological diseases. Abud et al. showed that human microglial-like cells could be differentiated from iPSCs. These cells secrete cytokines in response to inflammatory stimuli, migrate, and undergo calcium transients, and avidly phagocytose central nervous system substrates [80]. Human microglia-like cells derived from familial AD patients were also analysed, and the APOE4 genotype was found to profoundly impact several aspects of microglial function. This altered genotype impaired phagocytosis, migration, and metabolic activity but exacerbated cytokine secretion [43].

3D-differentiated neuronal cells expressing familial AD mutations also recapitulate Aβ- and tau-dependent pathology. This model could also facilitate the development of more precise human neural cell models of other neurodegenerative disorders [44]. Using brain organoids derived from AD patients, Raja et al. reported that iPSCs recapitulated AD-like characteristics, such as amyloid aggregation, hyperphosphorylated tau protein, and endosome abnormalities. Thus, they found that treating patient-derived organoids with β- and γ-secretase inhibitors significantly reduced the cellular phenotype associated with AD [29]. This 3D organoid system could also provide a platform for the development of new drug candidates for disease treatment.

#### 1.4.3. Amyotrophic Lateral Sclerosis

Amyotrophic lateral sclerosis (ALS) is a neurodegenerative disease characterized by the progressive degeneration of brain and spinal cord motor neurons. Its name reflects the degeneration of corticospinal motor neurons, as the descending axons in the lateral spinal cord seem scarred (lateral sclerosis), and there are diminished spinal motor neurons and muscle wasting (amyotrophy). Like most neurodegenerative diseases, it begins focally and subsequently spreads, with symptoms starting as subtle cramping or weakness in the limbs or bulbar muscles, progressing to paralysis of almost all skeletal muscles. Typically, death occurs three to five years after diagnosis [104].

Mutations in the TDP-43, *C9ORF72*, or SOD1 genes are most commonly related to familial ALS. Several groups have used iPSC-derived motor neurons, such as those harbouring mutations in TAR DNA binding protein-43 (TDP-43), to test drugs for treating familial forms of ALS. It was reported that iPSCs generated from an ALS patient differentiate into motor neurons harbouring mutations in TDP-43. In these samples, cytosolic aggregates formed similarly to those observed in postmortem tissue from ALS patients. Four chemical compounds were subsequently tested, and a histone acetyltransferase inhibitor, named anacardic acid, was found to rescue the abnormal ALS motor neuron phenotype. The authors suggested that anacardic acid may reverse ALS-associated phenotypes by downregulating TDP-43 mRNA expression [45]. In another study, fibroblast-derived iPSCs generated from healthy donors or patients with sporadic ALS were induced to differentiate into neurons. Only the neurons obtained from the patients exhibited the disease phenotype. It was shown that motor neurons derived from the iPSCs of three ALS patients had de novo TDP-43 aggregation and that the aggregates were similar to those observed in postmortem tissue. Using this model, the authors were able to demonstrate that several FDA-approved small molecules, including digoxin, could modulate TDP-43 aggregates [46]. In another study, motor neurons carrying a mutation in the C9ORF72 gene, one of the genes responsible for the disease, were generated. Expansions of a hexanucleotide repeat (GGGGCC) in the noncoding region of the C9ORF72 gene are the most common cause of the familial form of ALS. In this case, the neurons showed altered expression of genes involved in membrane excitability, including *DPP6*, demonstrating a diminished capacity to fire continuous spikes upon depolarization when compared to control motor neurons. Moreover, antisense oligonucleotides targeting the C9ORF72 transcript suppressed RNA focus formation and reversed the alterations in gene expression in the motor neurons [47]. RNA foci result from expanding RNA repeats, which are retained in the nucleus, assume unusual secondary folding, sequester some RNA-binding proteins, and can become toxic to the cell. SOD1 mutations induce a transcriptional signature characterized by increased oxidative stress, reduced mitochondrial function, altered subcellular transport, and activation of endoplasmic reticulum stress [48]. A system based on optimized all-optical electrophysiology for high-throughput functional characterization was used for testing drugs for disease phenotypes caused by a mutation in SOD1 [49].

Different neural cell types, such as astrocytes, oligodendrocytes, and motor neurons, contribute to ALS pathology; therefore, they should also be considered when developing reliable drug testing platforms. Several studies have indicated that astrocytes may mediate motor neuron death in this context. It has also been shown that mutations in genes that encode essential autophagy factors impair autophagy and may lead to neurodegenerative conditions such as ALS [105]. However, Madill et al. demonstrated that iPSCs generated from an ALS patient differentiated into astrocytes that modulated the autophagy pathway in a noncell autonomous manner. Data from this work suggested that patient astrocytes may modulate motor neuron cell death by impairing autophagic mechanisms [50]. Ferraiuolo et al. also showed that the death of motor neurons was induced by oligodendrocytes. In these cells, SOD1 was mutated in oligodendrocytes, which caused the death of control motor neurons and hyperexcitability when cocultured [51]. Therefore, the study of ALS with cells derived from iPSCs represents a new strategy for identifying effective drugs for treating this disease.

#### 1.4.4. Multiple Sclerosis

Multiple sclerosis (MS) is a chronic central nervous system inflammatory disease of autoimmune aetiology characterized by neuronal damage and axonal loss due to demyelination and subsequent degeneration. Activated T lymphocytes mediate this disease through the contributions of B lymphocytes and cells of the innate immune system [106]. Clinical symptoms of MS are variable and typically result from the involvement of sensory, motor, visual, and brainstem pathways and include fatigue, spasticity, and gait instability [106]. Researchers have developed a myelinating platform for drug screening using human pluripotent stem cells. Overexpression of the transcription factor SOX10 led to the generation of surface antigen O4-positive (O4+) and myelin basic protein-positive oligodendrocytes from pluripotent stem cells. Using this platform, the myelination of neurons by oligodendrocytes was demonstrated, and this platform could be applied for high-throughput screening to test the response to pro-myelinating drugs [52]. Researchers have also reported that, in an effort to discover compounds that increase the myelination of oligodendrocyte progenitor cells, a library of small bioactive molecules was screened on mouse pluripotent epiblast stem cell-derived oligodendrocyte progenitor cells. With this test, they found that two drugs, miconazole and clobetasol, were effective at promoting precocious myelination in organotypic cerebellar slice cultures and in vivo in early postnatal mouse pups. Furthermore, both drugs enhanced the generation of human oligodendrocytes from human oligodendrocyte progenitor cells in vitro [53]. These studies indicate that developing myelinating platforms for drug screening can lead to discoveries that can be translated into clinical practice.

iPSC-derived brain organoids have also been used for the study of MS. The authors reported deriving cerebral organoids from iPSCs of healthy control subjects as well as from primary progressive MS, secondary progressive MS and relapsing-remitting MS patients to understand the pathological basis of the varied clinical phenotypes of MS. In fact, most notably, in primary progressive MS, a decrease in the proliferation marker Ki67 and a reduction in the SOX2+ stem cell pool were observed, as was an increase in the expression of the neuronal markers CTIP2 and TBR1, as well as a strong decrease in oligodendrocyte differentiation. The brain organoids developed in this study from iPSCs from MS patients provide important information about the effect of patients’ genetic background on neural cells and interactions. This approach may provide novel insights into the development of the neural interactions that occur in MS patients [54].

#### 1.4.5. Huntington’s Disease

Huntington’s disease (HD) is an inherited neurodegenerative disorder characterized by neuropsychiatric symptoms, movement disability, and progressive cognitive impairment. Unfortunately, there is no effective therapy available for HD. The diagnosis is usually made by identifying an increased CAG repeat length in the huntingtin gene associated with clinical conditions. This genetic alteration results in the loss of GABAergic neurons in the striatum. At the cellular level, the mutation of the gene results in neuronal dysfunction and death through several mechanisms, including disruption of proteostasis, transcriptional and mitochondrial dysfunction, and direct toxicity of the mutant protein [107].

In a recent study, the authors reported the generation of iPSCs from HD patients and healthy individuals. A microarray profile distinguished the cell lines from healthy controls and patients, as the gene expression profile showed CAG repeat expansion. The iPSCs that differentiated into neural cells exhibited disease-associated differences in electrophysiology, metabolism, cell death, and longer CAG repeat expansions. This is because the severity of these disease-associated phenotypes is directly influenced by the extent of the CAG repeats. The strategy presented in that work provided a human stem cell platform for screening new therapeutic candidates for HD [55]. Using iPSCs from HD patients, Xu et al. reported a genetic correction using CRISPR-Cas9 that reversed the phenotypic abnormality [56]. The interaction between genome editing and iPSCs can expand the use of HD cellular models and therapeutic target discovery. Different disease phenotypes, such as aggregation of the mutated huntingtin protein, cell death, and neuronal toxicity, can be effective for drug testing [108].

In a disease model established from iPSCs, GABAergic medium spiny neurons were generated, which confirmed HD pathology in vitro, as evidenced by mutant huntingtin protein aggregation, an increased number of lysosomes/autophagosomes, increased nuclear indentations, and enhanced neuronal death during cell ageing. Furthermore, the drug EVP4593, a quinazoline derivative, reduced the number of lysosomes/autophagosomes and was neuroprotective during cell ageing. This approach provides a valuable tool for identifying candidate anti-HD drugs [57]. The severity of these disease-associated phenotypes is directly influenced by the extent of the CAG repeats [55]. In this context, researchers have also reported a model to investigate changes in the blood–brain barrier phenotype with the expansion of CAG repeats using an isogenic pair of iPSCs. These cells differentiate into brain microvascular endothelial-like cells, which, due to CAG expansion, exhibit subtle changes in phenotype, including differences in cell turnover and immune cell adhesion. The authors noted that the expansion of CAGs contributes to changes in the blood–brain barrier in HD [58]. Therefore, iPSC-derived cells provide a reliable model that allows drug testing and targeting of drugs to counteract HD pathology.

## 2. Conclusions

The development of iPSC technology represents a breakthrough in the medical field. The application of this technology to neurodegenerative diseases has already resulted in major advances in modelling this complex group of pathologies, revealing the related cellular and molecular mechanisms, as described here. The three major degenerative diseases, PD, ALS, and AD, are characterized by abnormal specific proteins inside and outside of neurons: TDP-43 in ALS, α-synuclein in PD, and tau and β-amyloid in AD. In addition to the abnormal aggregation of proteins, as shown in Figure 4, other common endophenotypes, such as reduced mitochondrial activity, accumulation of reactive oxygen species (ROS), and enhanced inflammation, are found in these diseases and can be investigated using iPSCs. The production of brain organoids formed by specialized neural cells derived from patient-specific iPSCs has accelerated the development of drug discovery platforms for treating neurodegenerative diseases, allowing the screening of new drugs, determining the significance of already commercialized drugs intended to treat other target diseases, and the development of neurodegenerative cell models. Both familial and sporadic phenotypes related to neurodegenerative diseases have been demonstrated using iPSCs derived from patients and their corresponding genetically edited isogenic cell lineage [19].

Despite these significant advances, several open pathways are still under investigation with the goal of developing a universal and robust neuronal platform derived from iPSCs for drug screening and for studying the pathogenesis of these diseases in vitro with human cells. Continuous developments are being made to address the technical obstacles, such as improving culture differentiation protocols aimed at increasing differentiation efficiency and cell maturation, combined with optimizing brain organoid technology [109,110]. Multiomics single-cell analysis may lead to the use of a new tool in the search for high-purity specific cell lineages that recapitulate the intended phenotype and underlying mechanism of these diseases in a reproducible, robust, and consistent way [20,111,112]. With the focus on mimicking a disease, researchers are already using the genetic induction of cellular ageing, genetic editing, and small molecules to reproduce the patient phenotype, including late-onset disease manifestations, familial or sporadic forms, and even environmental factors that could be important in this scenario [113]. By combining all of these initiatives with bioinformatics and computational and statistical analysis, a pattern of clinical and biological features is under construction, highlighting the promising role of iPSCs in drug discovery and neurodegenerative disease modelling.

## Figures and Tables

**Figure 1 ijms-25-02392-f001:**
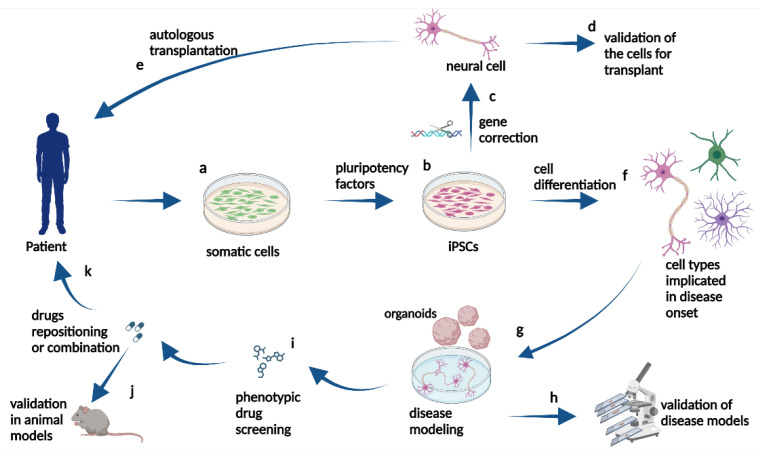
Schematic representation of iPSC applications. Somatic cells (**a**) can be obtained from patients with neurodegenerative diseases and induced to differentiate into iPSCs by pluripotency factors (**b**). Then, they can be genetically manipulated to undergo gene correction (**c**) and differentiated (**f**) into cell types implicated in disease onset. iPSCs can be differentiated into healthy neural cells for correct cell function and validation for transplant (**d**) or autologous transplantation in donor patients (**e**). Alternatively, genetically corrected iPSCs (**b**) can be differentiated into neural cells implicated in disease onset (**f**) to model the cellular pathogenic phenotype in vitro (**g**). These cellular cultures or organoids can be studied in the laboratory to validate iPSC-derived models (**h**) and can be used to screen drugs, for example (**i**). The selected drugs can be tested in animal models in preclinical trials (**j**), and in the case of beneficial results indicating drug repositioning or combination therapy (**k**), clinical trials in patients can be proposed.

**Figure 2 ijms-25-02392-f002:**
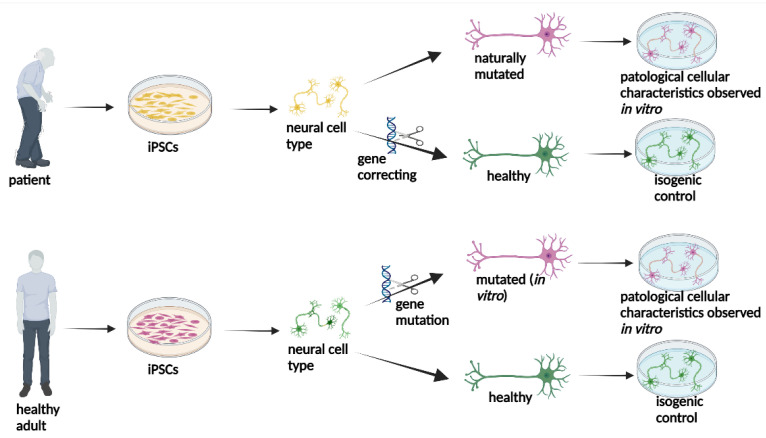
A representative figure of the potential of iPSCs obtained from a patient or a healthy adult is shown. The neural cells obtained from patients can be corrected by gene editing to obtain healthy neural cells, which can be used as an isogenic control in modelling this disease. Healthy neural cells obtained from a healthy adult can undergo gene editing, and neural cells harbouring a mutation from a neurodegenerative disease can be obtained for disease modelling.

**Figure 3 ijms-25-02392-f003:**
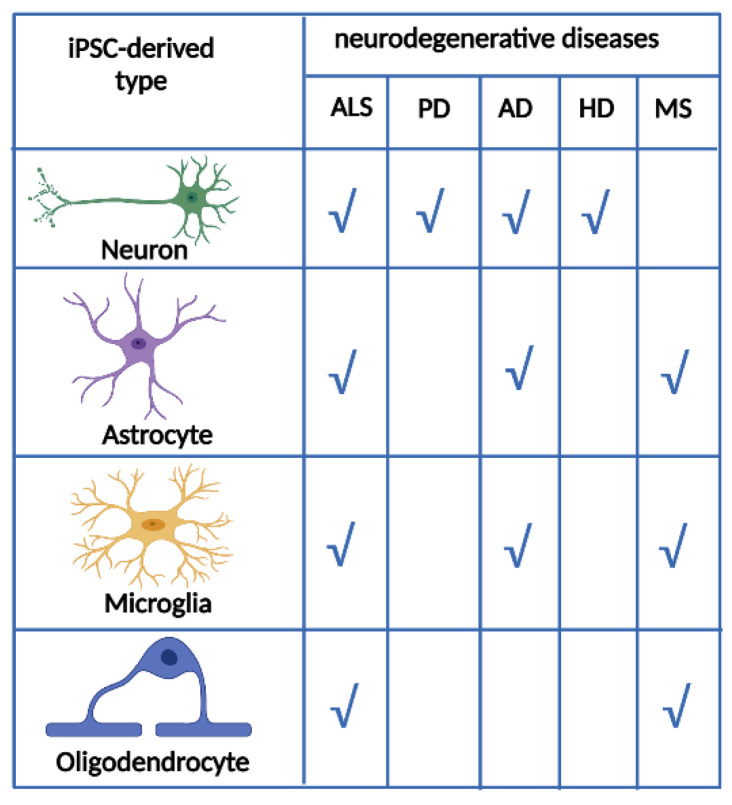
Table showing the relevance of each type of neural cell obtained from iPSCs for each neurodegenerative disease. ALS: amyotrophic lateral sclerosis, PD: Parkinson’s disease, AD: Alzheimer’s disease, HD: Huntington’s disease, MS: multiple sclerosis.

**Figure 4 ijms-25-02392-f004:**
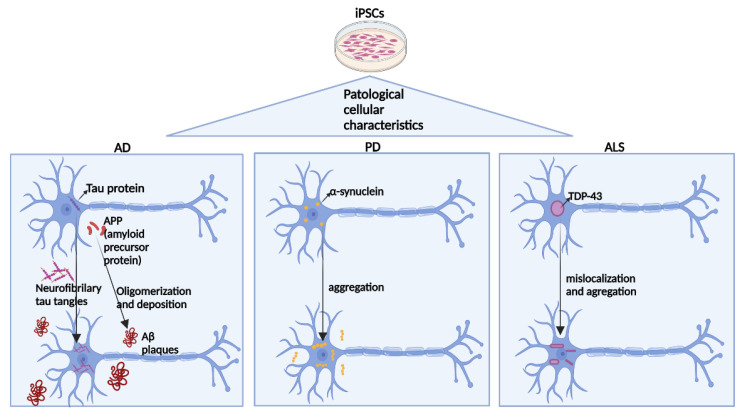
Common general pathogenic mechanisms of the three main neurodegenerative diseases can be obtained for disease modelling from iPSCs. In AD, the tau protein and amyloid precursor protein (APP) that is present in the healthy brain undergo aggregation during disease progression. The tau protein forms neurofibrillary tau tangles inside neurons, and the APP protein undergoes oligomerization and deposition, forming extracellular AB plaques. α-Synuclein present within cells in the healthy brain undergoes aggregation in PD patients. Finally, in ALS, the TDP-43 protein in the nucleus in the healthy brain migrates to the cytoplasm and undergoes aggregation during disease progression.

**Table 1 ijms-25-02392-t001:** iPSC lines created to study neurodegenerative diseases.

Disease	Gene Mutation	Phenotype	Cell Type	Potential Compound	References
PD	Triplication of SNCA	Increased α-synuclein	Dopaminergic neurons		[31]
PD	Heterozygous glucocerebrosidase mutation (GBA N370S)	∼50% glucocerebrosidase enzymatic activity, ∼3-fold elevated α-synuclein protein levels, and a reduced capacity to synthesize and release dopamine	Midbrain dopaminergic neurons	Monoamine oxidase B inhibitors	[32]
PD	Idiopathic Parkinson’s disease		Midbrain dopaminergic progenitor cells		[33]
AD	Presenilin 1 and presenilin 2	Increased amyloid β42 secretion	Neurons	γ-secretase inhibitors and modulators	[34]
AD	Amyloid precursor protein mutation (V717I)	Increased generation of both Aβ42 and Aβ38 and increase in levels of total and phosphorylated tau	Neurons		[35]
AD	G384A mutation of the presenilin 1 gene, which encodes presenilin-1	Increased the production of Aβ42 as a toxic Aβ species, and the Aβ42/40 ratio	Cortical neurons	Three candidates were combined to improve the anti-Aβ effect (bromocriptine, cromolyn, and topiramate) as an anti-Aβ cocktail	[36]
AD		Aβ1–42 aggregates	Neurons	Several small molecules as effective blockers against Aβ1-42 toxicity, including a Cdk2 inhibitor	[37]
AD	Duplication of the amyloid-β precursor protein-encoding gene	Significantly higher levels of the pathological markers amyloid-β, phospho-tau (Thr 231) and active glycogen synthase kinase-3β (aGSK-3β)	Neurons	β-secretase inhibitors	[38]
AD	MAPT gene	Abnormal tau expression, hyperphosphorylation of tau aggregates, and multiple disease phenotypes	Neurons		[39]
AD	Variations in amyloid precursor protein or presenilin 1 genes	Loss of synaptic proteins, increased ratio of intracellular and extracellular Aβ42/Aβ40 peptides, differences in protein aggregation, tau phosphorylation, miRNA pattern, and protein network alterations	Hippocampus neurons		[40]
AD		Mislocalization and abnormal expression of mature astrocyte markers, compromised astrocyte heterogeneity and astroglial atrophy.	Neurons and astrocytes		[41]
AD	Presenilin 1 ΔE9 mutation	Increased β-amyloid production, altered cytokine release, and dysregulated Ca^2+^ homeostasis.	Astrocytes		[42]
AD	APOE4 genotype	Impaired phagocytosis, migration, and metabolic activity but exacerbated cytokine secretion	Microglia-like cells		[43]
AD	FAD mutations in β-amyloid precursor protein and presenilin 1	Robust extracellular deposition of amyloid-β, including amyloid-β plaques. Silver-positive aggregates of phosphorylated tau in the soma and neurites, as well as filamentous tau.	3D-differentiated neuronal cells		[44]
AD	Amyloid precursor protein duplication or presenilin 1 mutation	Amyloid aggregation, hyperphosphorylated tau protein, and endosome abnormalities	Brain organoids	β- and γ-secretase inhibitors	[29]
ALS	TAR DNA binding protein-43 (TDP-43)	Cytosolic aggregates similar to those seen in postmortem tissue from ALS patients and exhibited shorter neurites as seen in a zebrafish model of ALS.	Motor neurons	Histone acetyltransferase inhibitor, named anacardic acid	[45]
ALS		TDP-43 aggregation	Motor neurons	FDA-approved small molecule modulators including digoxin	[46]
ALS	Mutation in the C9ORF72 gene	Altered expression of genes involved in membrane excitability, including *DPP6*, demonstrating a diminished capacity to fire continuous spikes upon depolarization compared to control motor neurons	Motor neurons	Antisense oligonucleotides targeting the C9ORF72 transcript suppressed RNA foci formation and reversed gene expression alterations in motor neurons	[47]
ALS	Mutant SOD1	Increased oxidative stress, reduced mitochondrial function, altered subcellular transport, and activation of the ER stress and unfolded protein response pathways	Motor neurons		[48]
ALS	Mutation SOD1 A4V	Elevated spike rates under weak or no stimulus and greater likelihood of entering depolarization block under strong optogenetic stimulus	Motor neurons		[49]
ALS		Increase in the expression of SOD1, a protein associated with the development of ALS. Astrocytes modulate the autophagy pathway in a noncell autonomous manner.	Astrocytes		[50]
ALS	Oligodendrocytes were mutated for SOD1	Death of motor neurons induced by oligodendrocytes	Oligodendrocytes		[51]
MS	Overexpression of the transcription factor SOX10	Using in vitro oligodendrocytes-neuron cocultures, myelination of neurons by oligodendrocytes were demonstrated	Oligodendrocytes		[52]
MS		Principal source of myelinating oligodendrocytes	Oligodendrocyte progenitor cells	Miconazole and clobetasol	[53]
MS		Decrease of proliferation marker Ki67 and a reduction of the SOX2+ stem cell pool associated with increased expression of neuronal markers CTIP2 and TBR1 as well as a strong decrease of oligodendrocyte differentiation	Brain organoids		[54]
HD	The microarray profile distinguished the lines of cells from healthy controls and patients as the gene expression profile showed CAG repeat expansion	Neural cells showed disease-associated differences in electrophysiology, metabolism, cell death, and longer CAG repeat expansions	Neural cells		[55]
HD	CAG repeat expansion in huntingtin gene. Genetic correction using CRISPR-Cas9	Impaired neural rosette formation, increased susceptibility to growth factor withdrawal, and deficits in mitochondrial respiration	Forebrain neurons		[56]
HD	Expansion of the CAG repeat in exon 1 of the huntingtin gene	Mutant huntingtin protein aggregation, increased number of lysosomes/autophagosomes, nuclear indentations, and enhanced neuronal death during cell ageing	GABAergic medium spiny neurons	EVP4593 drug, a quinazoline derivative	[57]
HD	CAG repeat expansion in huntingtin gene	Subtle changes in phenotype, including differences in cell turnover and immune cell adhesion	Brain microvascular endothelial-like cells		[58]

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
