# Peer review of "Induced Pluripotent Stem Cells in Drug Discovery and Neurodegenerative Disease Modelling"

_ijms, 2024, doi:10.3390/ijms25042392_

Round 1

Reviewer 1 Report

Comments and Suggestions for Authors

The manuscript by Beghini and colleagues summarizes the advances in iPSC technology in disease modeling and drug screening. Specifically, it focusses on neurodegenerative diseases. The authors should revise the English language of the review. I suggest having a native speaker read it.

The description of the iPSC model is in my opinion currently superfluous, so I would perhaps streamline that part and focus on the 3D models, by mentioning the different methods currently used to obtain brain organoids.

I would enrich the conclusions section with some future perspectives.

Comments on the Quality of English Language

The authors should revise the English language of the review. I suggest having a native speaker read it.

Author Response

Dear Editor,

We want to thank the reviewers for the thorough evaluation of our work and the constructive comments and suggestions. The detailed point-by-point reply to each reviewer’s concerns and criticisms is provided below.

# Reviewer 1

The authors should revise the English language of the review

  1. R. Dear editor, the manuscript was sent for full English language review by the American Journal Experts.

“The description of the iPSC model is in my opinion currently superfluous, so I would perhaps streamline that part and focus on the 3D models, by mentioning the different methods currently used to obtain brain organoids.”

  1. We thank the reviewer's suggestion and added a paragraph focusing on 3D brain organoids and the methods currently used to obtain these structures (lines 135 to 152).

Reviewer 2 Report

Comments and Suggestions for Authors

Beghini et al present a review on the use of iPSC technology in neural drug discovery and disease modeling. The review covering literature associated with application of iPSC technology to study or model Parkinson's disease, Alzheimer's disease, ALS, MS, and Huntington's disease. The review is compelling, however, would benefit from a few modifications listed below. 

1) The authors need to ensure the iPSC (singular), iPSC-derived and iPSCs (plural) are used correctly. This an issue that is seen throughout the manuscript in lines 19, 76, 80, 118, 130, 181, 189, 207, 210, 239, 248-251, 257, etc. 

2) The same goes for neural, neuronal, neuron, Parkinson’s, and Alzheimer's. For example, on line 32.

3) There are other grammatical issues throughout such as:

-line 55 should read "our brain is comprised of neurons.....

-line 62 stromal support and surround neurons

-line 76 allowed iPSC technology.....

-lines 111-113 and 123-125 should be rephrase

- line 267 repurposing.....

- among many others throughout the manuscript.

4) The authors should also elaborate either by dedicating a section or in more depth in the discussion the nature and limitations that result from the current immature neurons that are produced by all present differentiation studies and what is being done to address this issue. 

5) In line 141, the authors should cite relevant studies for example:

doi: 10.1371/journal.pone.0161969

doi: 10.1002/advs.202101462

doi: 10.1016/j.neuron.2019.07.010

doi: 10.1186/s13195-017-0234-1

6) In line 190, the authors should cite more relevant studies for example:

doi: 10.1016/j.stemcr.2023.06.007

doi: 10.1038/nbt.1877

doi: 10.1038/s41586-019-1289-x

Comments on the Quality of English Language

See comments and suggestions.

Author Response

Dear Editor,

We want to thank the reviewers for the thorough evaluation of our work and the constructive comments and suggestions. The detailed point-by-point reply to each reviewer’s concerns and criticisms is provided below.

# Reviewer 2

1) The authors need to ensure the iPSC (singular), iPSC-derived and iPSCs (plural) are used correctly. This an issue that is seen throughout the manuscript in lines 19, 76, 80, 118, 130, 181, 189, 207, 210, 239, 248-251, 257, etc.”

R.  Dear reviewer, we are sorry for this inconsistency, which was addressed in the revised version of the manuscript.

2) “The same goes for neural, neuronal, neuron, Parkinson’s, and Alzheimer's. For example, on line 32.”

R. We also ensured that these inconsistencies were appropriately addressed in the revised version.

3)There are other grammatical issues throughout such as:

-line 55 should read "our brain is comprised of neurons.....

-line 62 stromal support and surround neurons

-line 76 allowed iPSC technology.....

-lines 111-113 and 123-125 should be rephrase

- line 267 repurposing.....

- among many others throughout the manuscript.”

R. Dear editor, as already mentioned above for reviewer 1, to ensure that all issues were corrected, we submitted the entire manuscript for English language review by the American Journal experts. We believe that all grammatical issues were resolved.

4) “The authors should also elaborate either by dedicating a section or in more depth in the discussion the nature and limitations that result from the current immature neurons that are produced by all present differentiation studies and what is being done to address this issue.”

R. We thank the reviewer's suggestion and added a paragraph to the text discussing the limitation that results from obtaining immature neuronal cells in differentiation protocols. The added paragraph is found in the text between lines 286 and 308.

5) “ In line 141, the authors should cite relevant studies for example:

doi: 10.1371/journal.pone.0161969

doi: 10.1002/advs.202101462

doi: 10.1016/j.neuron.2019.07.010

doi: 10.1186/s13195-017-0234-1”

R. We agree with the studies indicated by the reviewer and added these references to the text that were cited in line 159.

6) “In line 190, the authors should cite more relevant studies for example:

doi: 10.1016/j.stemcr.2023.06.007

doi: 10.1038/nbt.1877

doi: 10.1038/s41586-019-1289-x”

R. The studies indicated by the reviewer were cited in line 217 to ensure that the reader has access to more relevant studies showing other subtypes of neurons and brain organoids that can be used in disease modeling and cell therapy.

Reviewer 3 Report

Comments and Suggestions for Authors

Dear editor,

The review is focused on the role of iPSCs in drug discovery and neurodegenerative disease modeling.

The MS is well written and summarizes the data on the topic. The main part of the MS is dedicated to examples of iPSC usage for modelling particular neurodegenerative diseases. The figures are well designed and helpful in understanding the material.

I’d like to suggest adding a Table with the most important references for modelling AD, PD and other conditions with iPSCs with short descriptions (cell type/organoid generation, type of genetic modification etc) for clarity. Now that information is inside the text and not easily accessible.

Overall, the data presented in the review is interesting. I’d recommend the article for publication after the minimal revision.

Minor remarks:

L13-14 “These adult cells are manipulated in vitro to express genes and factors essential for acquiring and maintaining embryonic stem cell (ESC) properties” The phrasing and meaning are unclear.

L19 and later throughout the text: “iPSC” should be written as “iPSCs” “cells” in plural are meant.

L66 same with “ESCs”

L75 The abbreviation “ES's” was not introduced and should be “ESCs”

L131 “techniques can create” – should be reformulated, “techniques” cannot create, researchers can.

L136 As far as I know there is no such word as “cytosystems”.

L208 “This protocol consisted of seven steps with an average duration of 150 days” – it seems from the wording that each step needs to last 150 days.

L575-… References are with double numbers.

Comments on the Quality of English Language

The language quality is basically good, some minor corections are necessary.

Author Response

Dear Editor,

We want to thank the reviewers for the thorough evaluation of our work and the constructive comments and suggestions. The detailed point-by-point reply to each reviewer’s concerns and criticisms is provided below.

# Reviewer 3

“I’d like to suggest adding a Table with the most important references for modelling AD, PD and other conditions with iPSCs with short descriptions (cell type/organoid generation, type of genetic modification etc) for clarity. Now that information is inside the text and not easily accessible.”

R. Dear editor, we thank the reviewer's suggestion and prepared a Table showing the IPSC lines generated to model the main neurodegenerative diseases, with the main mutations to model their phenotypes, the compounds used for testing, and the references. The table is inserted in the text between lines 186 and 187.

L13-14 “These adult cells are manipulated in vitro to express genes and factors essential for acquiring and maintaining embryonic stem cell (ESC) properties” The phrasing and meaning are unclear.”

R. We understand the reviewer's comment and, to make the language clearer, we submitted the manuscript to the American Journal Experts and obtained the language review certificate.

L19 and later throughout the text: “iPSC” should be written as “iPSCs” “cells” in plural are meant.

R. We made these necessary corrections.

“L66 same with “ESCs””

R. We also made this necessary correction.

“L75 The abbreviation “ES's” was not introduced and should be “ESCs””

R. We made this correction, which can be seen in line 83.

L131 “techniques can create” – should be reformulated, “techniques” cannot create, researchers can.”

R. The correction was made in line 154.

“L136 As far as I know there is no such word as “cytosystems”.”

R. Dear reviewer, the concept of cytosystems is given in the review published in 2013 - DOI: 10.1038/nature11859.

L208 “This protocol consisted of seven steps with an average duration of 150 days” – it seems from the wording that each step needs to last 150 days.”

R. We made the correction in lines 244 and 245 to make the sentence clearer.

“L575-… References are with double numbers.”

R. References have been appropriately corrected.

We hope the revised version of our work and this rebuttal will adequately address all concerns and invaluable suggestions. We believe that the present version is substantially improved.            
